# Identifiable Equivariant Networks are Layerwise Equivariant

**Vahid Shahverdi** [* 1]  **Giovanni Luca Marchetti** [* 1]  **Georg Bökman** [* 2]  **Kathlén Kohn** [* 1]

## Abstract

We investigate the relation between end-to-end equivariance and layerwise equivariance in deep neural networks. We prove the following: For a network whose end-to-end function is equivariant with respect to group actions on the input and output spaces, there is a parameter choice yielding the same end-to-end function such that its layers are equivariant with respect to some group actions on the latent spaces. Our result assumes that the parameters of the model are identifiable in an appropriate sense. This identifiability property has been established in the literature for a large class of networks, to which our results apply immediately, while it is conjectural for others. The theory we develop is grounded in an abstract formalism, and is therefore architecture-agnostic. Overall, our results provide a mathematical explanation for the emergence of equivariant structures in the weights of neural networks during training – a phenomenon that is consistently observed in practice.

## 1. Introduction

An important inductive bias in contemporary machine learning models is *equivariance* with respect to symmetries of data. It has played a pivotal role in several domains, underlying the success of convolutional neural networks in vision, graph neural networks in relational data, and equivariant transformers in physical and biological modelling. The design and analysis of equivariant neural networks lie at the heart of the discipline known as *geometric deep learning* (Bronstein et al., 2021).

A default paradigm in geometric deep learning is to design equivariant networks in a layerwise manner. It is typical to focus on equivariant linear maps, which are often feasible

---
[1]Department of Mathematics, KTH Royal Institute of Technology, Stockholm, Sweden [2]University of Amsterdam, The Netherlands. Correspondence to: Vahid Shahverdi <vahidsha@kth.se>.

*Proceedings of the 43rd International Conference on Machine Learning*, Seoul, South Korea. PMLR 306, 2026. Copyright 2026 by the author(s).

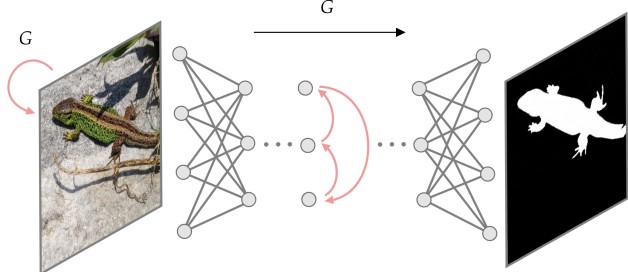

Figure 1. An image segmentation model is equivariant to rotations of the image. Our main result implies that the group action on the input propagates through the network via latent symmetries (e.g., neuron permutations), until it reaches the output.

to describe and parametrize. A deep network is then built from layers whose weights implement these maps. While this approach is effective, the extent of its generality is unclear. A major open question is how end-to-end equivariance can be engineered into a deep network without a layerwise approach. This question is closely linked to an active debate around the relation between (layerwise) equivariance and symmetry-based data augmentation (Brehmer et al., 2024; Nordenfors et al., 2023; Xie & Smidt, 2025). Indeed, it is empirically well supported in the literature that neural networks that are trained on data with a symmetry often encode this symmetry linearly in the latent spaces of the intermediate layers (Lenc & Vedaldi, 2015; Gruver et al., 2023; Bökman & Kahl, 2023). In other words, approximately equivariant neural networks are often approximately layerwise equivariant, but it remains unclear when this happens and why.

In this work, we address these questions theoretically by studying the structure of deep neural networks that are end-to-end equivariant with respect to groups of symmetries acting on the input and on the output space. We prove a somewhat surprising result: any such network *must be layerwise equivariant*, with respect to some group actions on the latent spaces. More precisely, we prove that the parameters of each layer must be equivariant, up to discarding 'inactive' neurons, which do not participate in the forward pass. This has two main consequences. First, designing equivariant layers is the only possible way to construct end-to-end equivariant networks, answering the above question. Second, if a deep network (with a non-equivariant architec-

ture) converges to an equivariant solution during training – due to either symmetries inherent in data or symmetry-based augmentation – then equivariance *emerges* automatically in the (active part of the) layers of the network.

The core assumption behind our results is parameter *identifiability* of the given model. Namely, we assume that parameters realizing the same function must be related by symmetries of the latent spaces. We require this to hold at least for parameters that do not come from smaller embedded architectures, i.e., from *submodels*. Functions induced by submodels are degenerate due to the presence of inactive neurons, and typically exhibit special symmetries. This identifiability property is a natural assumption that is known to be satisfied by several architectures. For Multilayer perceptrons (MLPs), it has been previously established for a variety of activation functions (e.g., Tanh, Sigmoid, powers). For ReLU, it is a well-known open problem, that is considered challenging due to certain pathological behavior.

We formulate our result in a highly-abstract language. Namely, deep models are abstracted as sequential compositions of parametric maps representing layers, and symmetries are abstracted as arbitrary groups acting on the latent spaces. As a consequence, the theory we develop is model-agnostic, and is applicable across different network architectures and symmetries. In fact, we discuss how our theory applies to both MLPs, and attention-based networks. Our proof strategy is related to – and actually generalizes – previous results from the literature (Agrawal & Ostrowski, 2022; Marchetti et al., 2024), that establish equivariant structures in the weights of specific invariant models, via reduction to identifiability. These works focus on shallow ReLU MLPs under a particular re-parametrization, and on the first layer of models with neuron-wise scaling symmetries, respectively.

In summary, our contributions are:

- We formulate identifiability, equivariance, and submodels, in an abstract language.

- Within this formalism, we prove that, under an identifiability assumption, equivariant networks are layerwise equivariant.

- We discuss how this result applies to MLPs and deep attention networks, both in theory, and in practice via an empirical investigation on image data.

## 2. Related Work

In this section, we review literature around equivariance and identifiability, as well as general mathematical perspectives that are relevant to our work.

**Identifiability.** Since our main result assumes parameter identifiability of the model, our work is closely related to, and builds upon, the line of literature concerning identifiability of deep neural networks. This is a classical question in the theory of deep learning, related to fundamental aspects such as sample complexity and interpretability. This research direction has recently seen a resurgence under the name of 'parameter symmetries' (Zhao et al., 2026; Lim et al., 2024).

Identifiability has been formally established for several instances of deep networks. For MLPs, the activation function plays a crucial role. Identifiability has been proven for sigmoidal activations (Fefferman et al., 1994; Vlačić & Bölcskei, 2022), for Tanh (Sussmann, 1992; Vlačić & Bölcskei, 2021), and for certain polynomial activations (Kileel et al., 2019; Finkel et al., 2025; Massarenti & Mella, 2025; Shahverdi et al., 2025b). For ReLU, it is known to be an extremely challenging problem, and only partial results are known (Grigsby et al., 2023; Petzka et al., 2020; Phuong & Lampert, 2020; Grillo & Montúfar, 2026). Regarding other architectures, some results are available for (linear) attention-based networks (Henry et al., 2025), and for (polynomial) convolutional ones (Shahverdi et al., 2025a; Hendi et al., 2026). We provide a more technical discussion of the identifiability results in Section 5.

**Algebro-geometric and Categorical Deep Learning.** Two recent lines of research, which are relevant to our work, propose to approach aspects of deep learning from the perspective of two related fields of pure mathematics – algebraic geometry and category theory, respectively.

The discipline known as *neuroalgebraic geometry* (Marchetti et al., 2025) has promoted the study of polynomial neural networks through the tools offered by algebraic geometry. These rich tools enable to formally establish sophisticated theoretical results in deep learning. Since fibers of polynomial maps are particularly well-behaved and tractable, a core focus of neuroalgebraic geometry is identifiability of polynomial models, such as MLPs with polynomial activation functions. In fact, several of the results mentioned in the previous section belong to this line of research.

Another recent discipline within theoretical deep learning has proposed to abstract deep learning models via the formalism of *category theory* (Gavranović et al., 2024; Fong et al., 2019). This enables to develop a general theory of models that is architecture-agnostic and compositional. In our work, we adopt, at least partially, the abstract formalism from categorical deep learning (see Remark 3.3), resulting in a theory that benefits from its generality.

**Equivariant Universal Approximation.** The mathematics behind equivariant neural networks has been extensively investigated, especially in the context of geometric deep learning. A large body of work focuses on *universal approximation* (Yarotsky, 2022; Pacini et al., 2025; Agrawal & Ostrowski, 2023; Ravanbakhsh, 2020; Petersen & Voigtlaender, 2020; Maron et al., 2019; Sonoda et al., 2022; Dym & Maron, 2020; Cen et al., 2026). These works establish that, under suitable conditions, layerwise equivariant architectures can approximate arbitrary continuous equivariant functions. Our main result is somehow orthogonal to this question. Rather than considering approximation properties of layerwise equivariant networks, we focus on their expressivity, showing that they can realize all deep equivariant networks. These two properties do not mathematically imply each other, unraveling different aspects of equivariant networks.

## 3. Formalism

In this section, we introduce the fundamental concepts behind the theory of this work. We define machine learning models in an abstracted formalism, which will enable us to prove our results in vast generality. For examples of the abstract definitions below, see Section 5 and Appendix B.

### 3.1. Models and Submodels

We start by introducing a general notion of a deep machine learning model.

**Definition 3.1.** A *model of depth* $L \in \mathbb{Z}_{\geq 1}$ is:

- A sequence of $L+1$ sets $V_0, \ldots, V_L$, representing latent spaces,

- A sequence of $L$ sets $\Theta_1, \ldots, \Theta_L$, representing parameter spaces (of intermediate layers),

- A sequence of $L$ maps $f_i \colon V_{i-1} \times \Theta_i \to V_i$, $i = 1, \ldots, L$, representing layers.

A deep model defines a map $f \colon V_0 \times \Theta \to V_L$ as the composition:

$$f(\bullet; \theta) = f_L(\bullet; \theta_L) \circ \cdots \circ f_1(\bullet; \theta_1), \quad (1)$$

where $\Theta = \Theta_1 \times \cdots \times \Theta_L$ is the total parameter space, and $\theta = (\theta_1, \ldots, \theta_L) \in \Theta$. In what follows, we will assume that all the (deep) models considered come from the same *class*, meaning that the sets and maps from Definition 3.1 belong to pre-scribed classes of sets and maps, respectively.

We will also need the notion of a submodel. As we shall argue in Section 5.1, this generalizes the notion of cloned and degenerate neurons in MLPs (Vlačić & Bölcskei, 2022).

**Definition 3.2.** Given a model of depth $L$ defined by $V_i, \Theta_i, f_i$, a *submodel* is a model of the same depth defined by $\widetilde{V}_i, \widetilde{\Theta}_i, \widetilde{f}_i$, equipped with maps $\alpha_i \colon \widetilde{V}_i \to V_i$, $\alpha_i^* \colon V_i \to \widetilde{V}_i$, $\beta_i \colon \widetilde{\Theta}_i \to \Theta_i$, such that:

- $\widetilde{V}_0 = V_0$, $\widetilde{V}_L = V_L$, and $\alpha_0, \alpha_0^*, \alpha_L, \alpha_L^*$ are identities maps,

- $\alpha_i^* \circ \alpha_i = \mathrm{Id}$ for $i = 0, \ldots, L$,

- the following diagram commutes for $i = 1, \ldots, L$:

$$
\begin{array}{ccc}
& V_{i-1} \times \Theta_i \xrightarrow{\ f_i\ } & V_i \\
\overset{\mathrm{Id} \times \beta_i}{\nearrow} & & \uparrow \alpha_i \\
V_{i-1} \times \widetilde{\Theta}_i & & \\
\underset{\alpha_{i-1}^* \times \mathrm{Id}}{\searrow} & \widetilde{V}_{i-1} \times \widetilde{\Theta}_i \xrightarrow{\ \widetilde{f}_i\ } & \widetilde{V}_i
\end{array}
$$

The commutative diagram formalizes the idea that, for parameters coming from the submodel, the layers of the model compute the same function as the ambient model, on all of their inputs. Again, we will assume that submodels belong to the same class as the original model. Clearly, each model is a submodel of itself, with $\alpha_i$, $\alpha_i^*$, $\beta_i$, all equal to identity maps – we refer to this as the *trivial* submodel. Moreover, the three conditions in Definition 3.2 imply that the end-to-end functions defined by a model and a submodel coincide: $f(x; \theta) = \widetilde{f}(x; \widetilde{\theta})$ for all $x \in V_0$ and $\theta \in \widetilde{\Theta}$, where $\theta_i = \beta_i(\widetilde{\theta}_i)$.

*Remark* 3.3. As anticipated in Section 2, Definition 3.1 and 3.2 can be interpreted in the language of category theory. A deep model is a sequence of composable morphisms in the 'category of parametric functions', and the maps $\alpha_i$ and $\beta_i$ of a submodel define a natural transformation between such sequences.

### 3.2. Symmetries and Identifiability

Throughout the work, we will deal with *symmetries* of models. Symmetries of a set $X$ are formalized via *group actions*, i.e. homomorphisms from a group $K$ to the group of bijections of $X$. We will denote group actions via $k \mapsto (x \mapsto k \cdot x)$, for $k \in K$ and $x \in X$. From now on, we will assume that all the latent spaces $V_i$ appearing in the models are equipped with an action by a group $K_i$. We denote this action, for $k \in K_i, x \in V_i$, as $(k, x) \mapsto k \cdot x$. Accordingly, we will assume that the maps $\alpha_i$ and $\alpha_i^*$ appearing in Definition 3.2 are compatible with the given symmetries. Formally, submodels are equipped with additional injective group homomorphisms $\gamma_i \colon \widetilde{K}_i \to K_i$, such that $\alpha_i$ and $\alpha_i^*$ are equivariant:

$$\alpha_i(\widetilde{k} \cdot \widetilde{x}) = \gamma_i(\widetilde{k}) \cdot \alpha_i(\widetilde{x}), \quad \widetilde{k} \cdot \alpha_i^*(x) = \alpha_i^*(\gamma_i(\widetilde{k}) \cdot x) \quad (2)$$

for all $\widetilde{x} \in \widetilde{V}_i, x \in V_i$ and $\widetilde{k} \in \widetilde{K}_i$. Lastly, we assume that $K_0$ and $K_L$ are the trivial group.

We now introduce a notion of identifiability for (deep) models. First, we wish to formalize the property that the only parameter symmetries arise from 'inter-layer transformations', i.e., by transforming the output of $f_i$ via the action of $K_i$ on $V_i$, and by cancelling the transformation back at the input of $f_{i+1}$. To this end, fix a model defined by $V_i, \Theta_i, f_i$.

**Definition 3.4.** A parameter $\theta \in \Theta$ is *identifiable* if for any $\theta' \in \Theta$ such that $f(\bullet; \theta) = f(\bullet; \theta')$, there exists a unique sequence of symmetries $k_i \in K_i$, $i = 0, \ldots, L$, such that for $i \geq 1$ and for $x \in V_{i-1}$:

$$f_i(x; \theta_i') = k_i \cdot f_i(k_{i-1}^{-1} \cdot x; \theta_i). \tag{3}$$

A subtlety is that we can not expect all the parameters of a model to be identifiable. Indeed, in most practical cases, parameters coming from submodels will exhibit 'degeneracy', resulting in additional symmetries. For several practical architectures, *almost all* parameters do not arise from submodels, and are in fact identifiable – see Section 5. However, we are interested in special parameters (i.e., corresponding to equivariant functions). In principle, all such parameters might come from submodels. Thus, we relax the definition of identifiability, requiring that the parameter is identifiable in the submodel it comes from.

**Definition 3.5.** A parameter $\theta \in \Theta$ is *weakly identifiable* if it is the image of an identifiable parameter of a submodel. Explicitly, there exists a submodel defined by $\widetilde{V}_i, \widetilde{\Theta}_i, \widetilde{f}_i$ and a parameter $\widetilde{\theta} \in \widetilde{\Theta}$ such that:

- $\widetilde{\theta}$ is identifiable for the submodel,

- $\theta_i = \beta_i(\widetilde{\theta}_i)$ for all $i = 1, \ldots, L$.

Note that every parameter is the image of itself from the trivial submodel. Thus, identifiable parameters are also weakly identifiable, motivating the terminology.

### 3.3. Equivariant Models

We now discuss the notion of end-to-end equivariance for machine learning models. Let $G$ be a group acting on both the input and output space $V_0, V_L$ of a given model.

**Definition 3.6.** A model is *G-equivariant* at $\theta \in \Theta$ if

$$f(g \cdot x; \theta) = g \cdot f(x; \theta) \tag{4}$$

for all $g \in G, x \in V_0$.

For our main result, we need to assume that the action is compatible with (the parameters of) the first and the last layer, in the following sense. The group $G$ also acts on $\Theta_1$ and $\Theta_L$, and these actions satisfy the *adjunction property*:

$$\begin{aligned} f_1(g \cdot x_0; \theta_1) &= f_1(x_0; g^{-1} \cdot \theta_1) \\ g \cdot f_L(x_{L-1}; \theta_L) &= f_L(x_{L-1}; g \cdot \theta_L) \end{aligned} \tag{5}$$

for all $g \in G, x_0 \in V_0, \theta_1 \in \Theta_1, x_{L-1} \in V_{L-1}, \theta_L \in \Theta_L$. We will assume that $G$ acts on these spaces also for submodels, and that the adjunction property holds as well.

## 4. Main Result

We now state our main result. Consider the setting of Section 3.3.

**Theorem 4.1.** *Let $\theta \in \Theta$ be a parameter. Suppose that:*

- *$\theta$ is weakly identifiable,*

- *the model is $G$-equivariant at $\theta$.*

*Then there exist group actions by $G$ on $V_i$, $i = 1, \ldots, L-1$, such that $f_i$ is $G$-equivariant at $\theta_i$, for every $i$.*

A proof is provided in the appendix, Section A.

*Remark* 4.2. It follows from the proof of Theorem 4.1 that, for $i = 1, \ldots, L$, the group action by $G$ on $V_i$ is actually induced by a group homomorphism $G \to K_i$. In other words, the latent action factors through the symmetry groups of the latent spaces.

*Remark* 4.3. For some common models, the adjunction property (5) does not hold, but there are group actions of $G$ in all layers that commute with the $K_i$ and such that the following generalized adjunction holds:

$$g^{-1} \cdot f_i(g \cdot x, \theta) = f_i(x, g^{-1} \cdot \theta) \tag{6}$$

for all $i = 1, \ldots, L, g \in G, x \in V_{i-1}, \theta \in \Theta_i$. In this case, Theorem 4.1 also holds with essentially the same proof. Interesting models include attention networks, where $G$ permutes tokens in all layers and acts trivially on the parameters, and convolutional networks on images, where $G$ rotates the input, yielding corresponding actions of $G$ on the latent spaces and the convolution filters $\theta$ (Bökman & Kahl, 2023). Lastly, for some architectures, even the generalized adjunction property (6) can fail. Examples include Deep Sets (Zaheer et al., 2017) and equivariant graph neural networks (Maron et al., 2018), where the (equivariant) linear layers can absorb specific group actions, but not arbitrary linear transformations. In these cases, our proof of Theorem 4.1 does not apply.

## 5. Examples

In this section, we explain how the abstract theory in Sec 3 concretizes for actual deep learning models.

### 5.1. Multi-Layer Perceptrons

A multi-layer perceptron (MLP) of depth $L \geq 1$ is a deep model specified by fixed widths $d_0, \ldots, d_L \in \mathbb{N}$ and an activation function $\sigma \colon \mathbb{R} \to \mathbb{R}$, and the sequences of spaces and maps as follows:

- The latent spaces $V_i$ are Euclidean spaces $\mathbb{R}^{d_i}$ for $i = 0, \ldots, L$.

- The parameter spaces $\Theta_i$ consist of weight matrices and bias vectors, i.e., $\Theta_i = \mathbb{R}^{d_i \times d_{i-1}} \times \mathbb{R}^{d_i}$.

- For each $\theta_i = (W_i, b_i) \in \Theta_i$, $i < L$, the layer map $f_i \colon V_{i-1} \times \Theta_i \to V_i$ is the composition of an affine map and the activation:

$$f_i(x; \theta_i) := \sigma(W_i x + b_i), \tag{7}$$

where $\sigma$ is applied component-wise. For $i = L$, we omit the activation: $f_L(x; \theta_L) := W_L x + b_L$.

When considering the class of MLPs, we fix $\sigma$, and let $d_0, \ldots, d_L$ vary. Moreover, we assume that $\sigma$ is origin-passing, i.e., $\sigma(0) = 0$.

**Symmetries.** We first discuss symmetries of MLPs. Since we wish (3) to hold, we set $K_i$ to be the *intertwiner group* of $\sigma$ (Godfrey et al., 2022), consisting of all invertible linear maps $A \in \mathrm{GL}(d_i)$ for which there exists $B \in \mathrm{GL}(d_i)$ such that

$$\sigma \circ A = B \circ \sigma. \tag{8}$$

Note that, since $\sigma$ is applied coordinate-wise, $K_i$ always contains permutation matrices. Moreover, it often happens that $K_i$ consists of *monomial matrices* (also referred to as generalized permutations), i.e., matrices that act on $\mathbb{R}^{d_i}$ by permuting and rescaling the coordinates. In other words, the symmetry group is contained in the semidirect product $(\mathbb{R}^{\times})^{d_i} \rtimes S_{d_i}$ between coordinate-wise rescalings and coordinate permutations. Intuitively, in this case latent symmetries are 'disentangled', meaning that they act neuron-wise. The allowed rescalings are determined by the homogeneity properties of $\sigma$. For power activations $\sigma(t) = t^m$ all rescalings are allowed, for ReLU only the positive ones, and for odd and even activations, such as Tanh, only sign flips $\{\pm 1\}$.

**Submodels.** We now discuss a natural construction of submodels of MLPs, where the maps $\alpha_i$ are linear. We represent each $\alpha_i$ by its associated $d_i \times \widetilde{d}_i$ matrix $A_i$. Moreover, denote by $(W_i, b_i)$ the image via $\beta_i$ of some parameter $(\widetilde{W}_i, \widetilde{b}_i) \in \widetilde{\Theta}_i$. For the diagrams in Definition 3.2 to commute for all submodel parameters, we require

$$A_i \sigma(\widetilde{W}_i A_{i-1}^* x + \widetilde{b}_i) = \sigma(W_i x + b_i), \tag{9}$$

where $A_{i-1}^*$ is the left inverse of $A_{i-1}$. A natural choice to satisfy this identity is to require $A_i$ to be a *rectangular intertwiner*, i.e., there exists a matrix $B_i \in \mathbb{R}^{d_i \times \widetilde{d}_i}$

such that $A_i \sigma(y) = \sigma(B_i y)$ for every $y \in \mathbb{R}^{\widetilde{d}_i}$. Substituting this into (9) and matching the pre-activation arguments, the parameter map $\beta_i \colon (\widetilde{W}_i, \widetilde{b}_i) \mapsto (W_i, b_i)$ satisfies $W_i = B_i \widetilde{W}_i A_{i-1}^*$ and $b_i = B_i \widetilde{b}_i$. Thus, submodels can be constructed, where $A_i$ is a rectangular intertwiner.

It is interesting to discuss in more detail the above-mentioned case when $K_i \subseteq (\mathbb{R}^{\times})^{d_i} \rtimes S_{d_i}$ consists of monomial matrices. In this case, intuitively, submodels of an MLPs correspond to parameters where some neurons are *inactive* – meaning that they do not participate in the forward pass – or *redundant* – meaning that they replicate other neurons in the same layer, up to an allowed rescaling. Formally, the matrix $A_i$ has, in every row, at most a single non-vanishing entry coming from an allowed scalar. Vanishing rows correspond to inactive neurons in $V_i$, while two rows with non-vanishing entries at the same index correspond to redundant ones. When a neuron is inactive, the corresponding column of $W_{i+1}$ and row of $W_i$ vanish, while for a pair of redundant neurons, the corresponding rows of $W_i$ are proportional.

**Identifiability.** With the above setting, the core question is whether parameters of MLPs are weakly identifiable (Definition 3.5), since this is the assumption of our main result (Theorem 4.1). For arbitrary activation functions, this is a major open problem in the theory of deep learning:

**Conjecture 5.1.** *For a broad class of activation functions $\sigma$, every function realized by an MLP with activation $\sigma$ admits a weakly identifiable representative. More precisely, for every parameter $\theta \in \Theta$, there exists a parameter $\theta' \in \Theta$ such that $f(\bullet; \theta) = f(\bullet; \theta')$, and $\theta'$ is weakly identifiable.*

*Remark* 5.2. The reason we need to consider $\theta'$ in Conjecture 5.1 is the following. A neuron in $V_i$ can be inactive when the corresponding column of $W_{i+1}$ vanishes, but the corresponding row of $W_i$ does not, or vice versa (and similarly for $b_i$). These parameters are not identifiable, and do not come from subnetworks, since the diagram in Definition 3.2 fails to commute. However, one can construct a parameter $\theta'$ by zeroing out the non-vanishing row/column, without changing the function $f(\bullet; \theta)$, and obtaining a submodel. We provide an illustration of this in the appendix, Section B.

This conjecture has been proven in several instances (here, all submodels are linearly embedded as in the construction above):

- *Sigmoidal family*: $\sigma$ belongs to a family of activations that is dense in an opportune space of piece-wise $C^1$ functions (Vlačić & Bölcskei, 2022; Fefferman et al., 1994). In other words, the result holds for a family of activations that can approximate arbitrary ones in the given function space. In this case, $K_i$ consists only of

permutations.

- *Large powers*: $\sigma(t) = t^m$ is a monomial of degree $m$ that is large with respect to $d_0, \ldots, d_L$ (Kileel et al., 2019; Finkel et al., 2025). In this case, $K_i$ consists of all monomial matrices.

- *Hyperbolic tangent*: $\sigma(t) = \text{Tanh}(t)$ (Sussmann, 1992; Vlačić & Bölcskei, 2021). Here, $K_i = \{\pm 1\}^{d_i} \rtimes S_{d_i}$ consists of permutations and sign flips.

- *Linear*: $\sigma(t) = t$ is the identity, and $d_1 = \cdots = d_{L-1}$ (Trager et al., 2020). In this case, $K_i = \text{GL}(d_i)$.

For deep ReLU-networks, identifiability is known to be a particularly challenging problem, and only partial results are available (Grigsby et al., 2023; Petzka et al., 2020; Phuong & Lampert, 2020; Grillo & Montúfar, 2026). One of the obstructions to identifiability is the relation $\sigma(x) - \sigma(-x) = x$ (also satisfied by for instance GELU), allowing specific parameter configurations to bypass the nonlinearity. This suggests that, for ReLU MLPs, a version of Conjecture 5.1 should be understood with some care, that is, the symmetry groups $K_i$ and the relevant submodels may have to be taken larger, a priori, than the standard positive rescalings and permutations in order to account for such degenerate regions. However, for shallow ReLU-networks, a re-parameterization of the network recovers identifiability and therefore layerwise equivariance of invariant networks (Agrawal & Ostrowski, 2022).

**Equivariance.** Finally, regarding end-to-end equivariance of MLPs, we remark that in order for the adjunction property (5) to hold, $G$ must act linearly on $V_0$ and $V_L$, i.e., the input and output space must define a group representation. In this case, the corresponding action on the first-layer parameters $\Theta_1$ is, for $A \in G \subseteq \text{GL}(d_0)$, $A \cdot (W_1, b_1) = (W_1 A^{-1}, b_1)$ and on the last-layer parameters $\Theta_L$ it is $A \cdot (W_L, b_L) = (AW_L, Ab_L)$.

**Corollary 5.3** (of Conjecture 5.1). *For a broad class of activation functions $\sigma$, every parameter $\theta \in \Theta$ of an MLP for which $f(\bullet; \theta)$ is equivariant under some group action admits an equivalent parameter $\theta' \in \Theta$ such that $f(\bullet; \theta) = f(\bullet; \theta')$ and each layer $f_i$ is equivariant at $\theta'_i$.*

### 5.2. Multi-Head Attention Networks

We now extend our formalism to multi-head attention models. We consider a simplified Transformer-like architecture of depth $L \geq 1$ consisting of pure attention layers without skip connections or layer normalization. Moreover, instead of parametrizing attention mechanisms via query and key matrices $W_Q$, $W_K$, we directly deploy the *attention matrix* $W_A := W_Q W_K^\top$. This simplifies the parametrization, and avoids dealing with the intra-layer symmetries between

keys and queries, which are irrelevant for our purposes. Lastly, we follow a non-standard convention in choosing the latent space. We consider as latents the outputs of the attention heads, before they are aggregated into tokens. This is convenient, since it will enable head permutations to be formalized as symmetries of the latent spaces. More precisely:

- The latent spaces $V_i$ represent, for each of the $h_i$ heads, sequences of $n$ vectors in $\mathbb{R}^{d_i}$, i.e., $V_i = \mathbb{R}^{n \times d_i \times h_i}$ (with $h_0 = h_L = 1$).

- The parameter space $\Theta_i = \mathbb{R}^{h_i \times d_{i-1} \times d_{i-1}} \oplus \mathbb{R}^{h_i \times d_{i-1} \times d_i} \oplus \mathbb{R}^{h_{i-1} d_{i-1} \times d_{i-1}}$ consists of head-wise attention and value matrices $W_A^{i,j}$ and $W_V^{i,j}$, $j = 1, \ldots, h_i$, and a cross-head projection matrix $W_O^i$.

- For each $\theta_i = (W_A^i, W_V^i, W_O^i) \in \Theta_i$, the layer map is defined for $j = 1, \ldots, h_i$ by:

$$f_i(X; \theta_i)[\bullet, \bullet, j] = \text{smax}\left(\overline{X} W_A^{i,j} \overline{X}^\top\right) \overline{X} W_V^{i,j}. \tag{10}$$

Here, smax denotes the softmax operator applied row-wise, and $\overline{X} = X W_O^i$ with $X \in \mathbb{R}^{n \times d_{i-1} h_{i-1}}$, i.e., the last two indices have been flattened.

**Symmetries, submodels, and identifiability** There are two natural sources of symmetries in multi-head attention. First, heads can be permuted. Second, since values depend linearly on the input, the token space $\mathbb{R}^{d_i}$ of each head can be transformed linearly. Thus, we set $K_i = \text{GL}(d_i)^{h_i} \rtimes S_{h_i}$. Given $(X[p, q, s])_{p,q,s} \in V_i = \mathbb{R}^{n \times d_i \times h_i}$, $S_{h_i}$ permutes the $s$ index of $X$, while the $q$ index gets contracted via left multiplication by a matrix in the $s$-th copy of $\text{GL}(d_i)$.

Intuitively, multi-head attention networks behave similarly to MLPs in the 'disentangled' case, i.e., when symmetries consist of monomial matrices. Each head behaves like a (sequence-valued) 'neuron'. The crucial difference is that in MLPs, the per-neuron symmetries are just scalings in $\mathbb{R}^\times$, while for attention, they can be arbitrary invertible transformations of the output space $\mathbb{R}^{d_i}$ of the corresponding head. Indeed, submodels can be constructed similarly to Section 5.1, resulting in parameters where some heads are redundant – i.e., implement the same function – or have low-rank structure – i.e., the output tokens live in a low-dimensional subspace of $\mathbb{R}^{d_i}$.

As with MLPs, identifiability of deep attention networks is not established in general. It is expected, analogously to Conjecture 5.1, that every parameter is weakly identifiable, up to replacing it with another one defining the same function. Recently, this has been established by Henry et al.

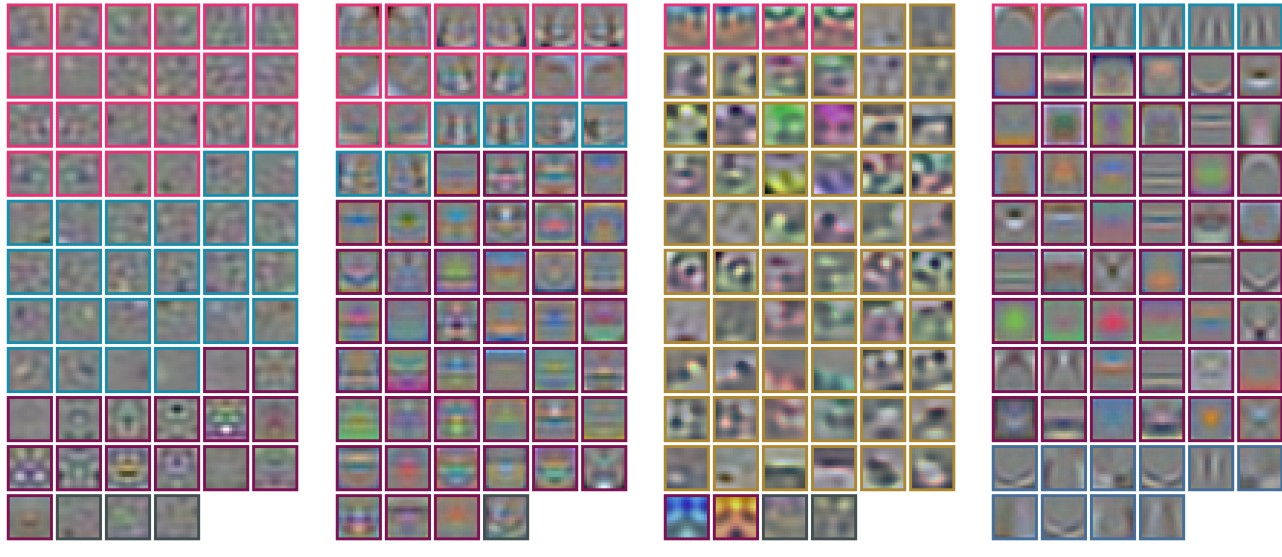

*(a)* Autoencoder with Tanh.      *(b)* Classifier with Tanh.      *(c)* Autoencoder with GELU.      *(d)* Classifier with GELU.

*Figure 2.* First-layer weights of MLPs trained on CIFAR10. Each square is a filter that maps an input RGB image to a single neuron of the subsequent layer. The filters have been sorted for illustrative purposes, with different filter categories highlighted by different colors.

**Pink:** Filters with a left-to-right mirrored copy (shown adjacent).
**Light Blue:** Filters with a left-to-right mirrored negated copy (shown adjacent).
**Gold:** Filters with a negated copy (shown adjacent).
**Purple:** Left-to-right symmetric.
**Dark Blue:** Left-to-right anti-symmetric.
**Gray:** Filters that do not fit into the other categories.

(2025) for a single-head model implementing *linear* attention, meaning that the softmax operator is removed from (10). For non-linear attention, identifiability of deep models is an open problem.

**Equivariance**   Again, the adjunction property (5) forces constraints on the allowed group actions on the input and output spaces. Similarly to MLPs, $G$ must act linearly on $V_0 = \mathbb{R}^{n \times d_0}$ and $V_L = \mathbb{R}^{n \times d_L}$. Moreover, since the attention and value matrices are applied token-wise, $G$ must act on all the tokens in the same way. Namely, $G$ acts on the token spaces $\mathbb{R}^{d_0}$ and $\mathbb{R}^{d_L}$, and this action is extended diagonally to sequences $X$ as $(g \cdot X)[p, \bullet] = g \cdot X[p, \bullet]$.

This type of group action does not cover all symmetries encountered in practice, since cross-token symmetries are common in many domains. An example is computer vision, where tokens correspond to image patches, and where symmetries (e.g., image rotations or reflections) typically permute tokens, in addition to transforming them. Token permutations can be described by the more general adjunction property in (6) and attention layers are a priori equivariant under token permutations. However, usually not all token permutations correspond to correct symmetries of the task at hand (e.g., permutations that are not rotations or reflections of images). A common solution to break incorrect symmetries is adding learnable absolute positional encod-

ings. With positional encodings, the standard adjunction property (5) holds so that equivariance under $G \subseteq S_n$ of an identifiable model implies equivariance of the positional encodings by our main result.

## 6. Illustrative Experiments

We now turn to illustrations of the theoretical results using small neural networks trained on CIFAR10 (Krizhevsky, 2009). The aim of this section is not to provide a large amount of quantitative results on learned equivariance in neural networks – which already exist to some extent in the literature (Lenc & Vedaldi, 2015; Gruver et al., 2023; Bökman & Kahl, 2023) – but to demonstrate the practical relevance of the theory, and to help build intuition around it. Unlike our theory, which assumes exact equivariance, trained networks are, naturally, only approximately equivariant. Thus, the purpose of our empirical investigation is to show that the layerwise structures predicted by the exact theory still emerges in practical approximate settings. We will look at both MLPs and multi-head attention layers.

### 6.1. Multi-Layer Perceptrons

We train small MLPs of depth $4$ on CIFAR10. Between each pair of linear layers there is a nonlinearity, where the last two nonlinearities are always pointwise GELU, while

the first nonlinearity is either pointwise Tanh or pointwise GELU. We consider two tasks – autoencoding and classification – differing by the degree of symmetry present on the output. Loosely speaking, a perfect autoencoder is expected to be equivariant under any transformation that maps a CIFAR10 image to a new image that looks like a CIFAR10 image. On the other hand, a perfect classifier is invariant under class-preserving image transformations. We focus on the simplest symmetry that is class-preserving, namely left-to-right mirroring.

We train MLPs using mean-squared loss for autoencoding and cross-entropy loss for classification. During the second half of the training we add an equivariance loss. For autoencoding the equivariance loss is a mean-squared loss between $f(x)$ and $\texttt{mirror}[f(\texttt{mirror}[x])]$ and for classification it is a mean-squared loss between $f(x)$ and $f(\texttt{mirror}[x])$, where $f$ is the neural network, $x$ is the input image and $\texttt{mirror}$ is the function that mirrors images left-to-right. This loss does not impose an exact equivariance constraint on the model, nor does it impose any layerwise structure. It only encourages the learned function to become approximately equivariant on the data distribution. Further details on the experimental setup are presented in Appendix C. We first study the trained networks qualitatively by looking at the 64 filters learned in the first layer, as shown in Figure 2.

In Figures 2a and 2b, we see that the networks trained with a Tanh nonlinearity encode the left-to-right mirroring symmetry by using a combination of symmetric filters and filters that have either a mirrored copy or a mirrored and negated copy. This means that when the input is mirrored, this corresponds to a signed permutation matrix acting on the latent space following the first layer. Signed permutation matrices precisely constitute the intertwiner group of Tanh, so here we see a clear example of identifiability leading to layerwise equivariance.

In Figures 2c and 2d, we see that using GELU leads to degenerate parameters, in particular for the autoencoder. The intertwiner group of GELU consists of permutation matrices only, but the relation $\sigma(x) - \sigma(-x) = x$ enables the MLP layers to bypass the nonlinearity and hence be equivariant without a permutation action on the latent space. The large amount of filters with negated copies (highlighted in gold) can thus be explained by the fact that the network is trained to approximate the identity on CIFAR10, which makes it beneficial to bypass the nonlinearity.

Next, we study the networks quantitatively. Letting $f_1$ be the first linear layer and the first nonlinearity and $f_2$ the first linear layer, the first nonlinearity and the second linear layer, we estimate linear transformations $A_i$ such that $A_i f_i(x) \approx f_i(\texttt{mirror}[x])$. This is done by least squares estimation, where $x$ consists of 100k noise

*Table 1.* Relative equivariance errors for the first layers of the networks visualized in Figure 2.

|     | Mirroring | Autoencoder | | Classifier | |
| --- | --- | --- | --- | --- | --- |
|     |           | Tanh | GELU | Tanh | GELU |
| $f_1$ | left-right | 0.029 | 0.40 | 0.077 | 0.19 |
| $f_1$ | up-down | 0.15 | 0.49 | 0.48 | 0.56 |
| $f_2$ | left-right | 0.022 | 0.11 | 0.054 | 0.064 |
| $f_2$ | up-down | 0.13 | 0.18 | 0.38 | 0.42 |

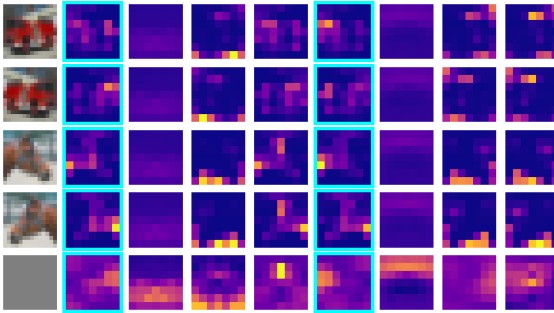

*Figure 3.* Visualization of a multi-head attention layer in a neural network autoencoder trained on CIFAR10. The left-most column shows the input image, and the remaining eight columns visualize the attention matrices (for the 'class token') of the heads. The two columns in cyan correspond to attention heads that permute when the input image is left-to-right mirrored.

samples. We compute the relative equivariance error $|f_i(\texttt{mirror}[y]) - A_i f_i(y)|/|f_i(y)|$ on 100k independent noise samples $y$. We include both left-right mirroring, which is what the networks are trained to be equivariant to, and up-down mirroring as a reference. In Table 1 we report the median error for the different cases.

Tanh networks are linearly equivariant both for $f_1$ and $f_2$, whereas GELU networks are much more linearly equivariant for $f_2$. This is again explained by the discussed identity $\sigma(x) - \sigma(-x) = x$ which requires two linear layers to realize. Left-right equivariance is stronger than up-down equivariance in all cases, although the autoencoders do exhibit a degree of up-down equivariance even without being trained for it. The autoencoding task in itself is equivariant to up-down mirroring, whereas this is more unclear for classification.

### 6.2. Multi-Head Attention

Next, we train an autoencoder with a multi-head attention layer on CIFAR10. The network is identical to the MLPs in the previous section except that we replace the first linear layer by a patch embedding followed by an attention layer. The attention implementation is inspired by common practice in vision transformers (Dosovitskiy et al., 2021).

More explicitly, we first use a stride 2 convolution with

filter size 2 to patchify the image. After the convolution, we interpret the feature vectors at the obtained subsampled pixel locations as 'image tokens'. Two different convolutions of the input yield key and value tokens, to both of which we add learnable positional encodings. We use a single learnable query token, which can be thought of as the class token in a vision transformer. The output of the attention layer is the sum over value tokens weighted by the softmax-attention from the single query token to each key token. We use multi-head attention with eight heads, so the above is carried out in parallel eight times (with eight different sets of network weights) to obtain the final output as the concatenation of the outputs of the different heads. Hence, the layer maps the entire image to a single token which is then processed using an MLP as in the preceding section. Detailed info on the network training is given in Appendix C.

Figure 3 shows the attention from the single query token to each key token across the heads of the multi-head attention layer. We observe that when the input image is left-to-right mirrored, the attention pattern of most heads is approximately mirrored. However, the highlighted first and fifth heads also permute with each other. This is a qualitative example of a multi-head attention layer that encodes mirror equivariance as a permutation over the heads.

## 7. Conclusions, Limitations, and Future Work

We have proven that, for an identifiable deep machine learning model, end-to-end equivariance with respect to group actions on the input and on the output spaces implies layerwise equivariance with respect to some group actions on its latent spaces. While the result is general and architecture-agnostic, several questions are left open.

The precise conditions for identifiability for networks with ReLU-style nonlinearities remains an important topic for future work, which using our framework directly translates to new results on layerwise equivariance in such networks.

Further, we have not covered skip connections, which are a common component of modern neural networks. Skip connections reduce the identifiability of networks due to increased inter-layer dependencies (for instance, zeroing out residual blocks alters the effective depth of the network). For an interesting account of skip connections in equivariant networks see (Agrawal & Ostrowski, 2023), where equivariant ReLU-networks are built using dependencies between the weights in all layers.

Finally, while our theory proves that weakly identifiable neural networks are layerwise equivariant, the results do not say what group actions on the intermediate latent spaces are obtained in practice, or what choice of group actions is optimal for approximating a given function. The result

is also existential, and does not explain how a layerwise equivariant parametrization may emerge during optimization in an unconstrained model. Our results also do not indicate whether it is better in practice to learn equivariance from data or constrain the network layers to be equivariant a priori. Constraining the network layers a priori can be fruitful, e.g., for efficiency in terms of parameters, data, and compute.

## Acknowledgements

This work was partially supported by the Wallenberg AI, Autonomous Systems and Software Program (WASP) funded by the Knut and Alice Wallenberg Foundation.

## Impact Statement

This paper presents work whose goal is to advance the field of machine learning. There are many potential societal consequences of our work, none of which we feel must be specifically highlighted here.

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

# A. Proof of the Main Result

Here, we provide the proof of Theorem 4.1.

*Proof.* Since $\theta$ is weakly identifiable, there exists a submodel defined by $\widetilde{V}_i, \widetilde{\Theta}_i, \widetilde{f}_i$ such that $\theta_i = \beta_i(\widetilde{\theta}_i)$ for some identifiable parameter $\widetilde{\theta} \in \widetilde{\Theta}$. Now, as remarked after Definition 3.2, it follows that $f(\bullet; \theta) = \widetilde{f}(\bullet; \widetilde{\theta})$. Since the model is $G$-equivariant at $\theta$, the submodel is $G$-equivariant at $\widetilde{\theta}$ as well. By the adjunction property applied to the submodel, we know that:

$$\widetilde{f}_1(g \cdot x; \widetilde{\theta}_1) = \widetilde{f}_1(x; g^{-1} \cdot \widetilde{\theta}_1) \tag{11}$$

for all $g \in G, x \in V_0$, and similarly for $\widetilde{f}_L$.

Now, fix $g \in G$. Define a new parameter for the submodel as:

$$\widetilde{\theta}_i' = \begin{cases} g^{-1} \cdot \widetilde{\theta}_i & i = 1, L, \\ \widetilde{\theta}_i & i \neq 1, L. \end{cases} \tag{12}$$

From $G$-equivariance of the submodel and (11), it follows that $\widetilde{f}(\bullet; \widetilde{\theta}) = \widetilde{f}(\bullet; \widetilde{\theta}')$. Since $\widetilde{\theta}$ is identifiable, there exists a unique sequence of symmetries $\widetilde{k}_i \in \widetilde{K}_i, i = 1, \ldots, L$, depending on $g$, such that:

$$\widetilde{k}_i \cdot \widetilde{f}_i(x; \widetilde{\theta}_i) = \widetilde{f}_i(\widetilde{k}_{i-1} \cdot x; \widetilde{\theta}_i') \tag{13}$$

for all $x \in \widetilde{V}_{i-1}$. We wish to show that, for every $i = 1, \ldots, L - 1$, the map $\rho_i \colon G \to \widetilde{K}_i$, $g \mapsto \widetilde{k}_i$, is a group homomorphism. Wet set $\rho_0$ and $\rho_L$ to the identity map $G \to G$, and proceed by induction on $i$. Given any $g \in G$, (13) implies that:

$$\rho_i(g) \cdot \widetilde{f}_i(x; \widetilde{\theta}_i) = \widetilde{f}_i(\rho_{i-1}(g) \cdot x; \widetilde{\theta}_i), \tag{14}$$

for all $x \in \widetilde{V}_{i-1}$, where we use adjunction (11) in the edge cases $i = 1, L$. In particular, given $g, h \in G$, (14) holds for $g$, $h$, and $gh$. Since, by the inductive hypothesis, $\rho_{i-1}$ is a homomorphism, we conclude that:

$$\begin{aligned} \rho_i(gh) \cdot \widetilde{f}_i(x; \widetilde{\theta}_i) &= \widetilde{f}_i(\rho_{i-1}(gh) \cdot x; \widetilde{\theta}_i) \\ &= \widetilde{f}_i(\rho_{i-1}(g) \cdot (\rho_{i-1}(h) \cdot x); \widetilde{\theta}_i) \\ &= \rho_i(g) \cdot \widetilde{f}_i(\rho_{i-1}(h) \cdot x; \widetilde{\theta}_i) \\ &= (\rho_i(g)\rho_i(h)) \cdot \widetilde{f}_i(x; \widetilde{\theta}_i). \end{aligned} \tag{15}$$

From the uniqueness of the symmetries $\widetilde{k}_i$, we deduce that $\rho_i(gh) = \rho_i(g)\rho_i(h)$, as desired.

Since $\rho_i$ is a homomorphism and $\widetilde{K}_i$ acts on $\widetilde{V}_i$, we obtain an induced action by $G$ on $\widetilde{V}_i$. The layers $\widetilde{f}_i(\bullet; \widetilde{\theta}_i)$ of the submodel are $G$-equivariant with respect to these actions by (13). By composing $\rho_i$ with the group homomorphism $\gamma_i \colon \widetilde{K}_i \to K_i$ (see Section 3.2), $G$ acts on $V_i$ as well. $G$-equivariance of the layers $f_i(\bullet; \theta_i)$ now follows from Definition 3.2 and the equivariance properties in (2), concluding the proof:

$$\begin{aligned} f_i(\gamma_{i-1}(\rho_{i-1}(g)) \cdot x; \theta_i) &= \alpha_i(\widetilde{f}_i(\alpha_{i-1}^*(\gamma_{i-1}(\rho_{i-1}(g)) \cdot x); \widetilde{\theta}_i)) \\ &= \alpha_i(\widetilde{f}_i(\rho_{i-1}(g) \cdot \alpha_{i-1}^*(x); \widetilde{\theta}_i)) \\ &= \alpha_i(\rho_i(g) \cdot \widetilde{f}_i(\alpha_{i-1}^*(x); \widetilde{\theta}_i)) \\ &= \gamma_i(\rho_i(g)) \cdot \alpha_i(\widetilde{f}_i(\alpha_{i-1}^*(x); \widetilde{\theta}_i)) \\ &= \gamma_i(\rho_i(g)) \cdot f_i(x; \theta_i). \end{aligned} \tag{16}$$

$\square$

## B. A Small Example of a Submodel

Consider a shallow network with the following end-to-end function parameterization:

$$f(x; \theta) = \begin{bmatrix} e & f \end{bmatrix} \sigma \left( \begin{bmatrix} a & b \\ c & d \end{bmatrix} \begin{bmatrix} x_1 \\ x_2 \end{bmatrix} \right). \tag{17}$$

For a sufficiently generic activation function as discussed in Section 5.1, the intertwiner group $K_1$ is isomorphic to the symmetric group $S_2$, generated by the permutation matrix $P = \begin{bmatrix} 0 & 1 \\ 1 & 0 \end{bmatrix}$. To be precise, for any parameter tuple $\theta = (W_1, W_2)$, the transformed parameter $\theta' = (PW_1, W_2P^{-1})$ yields the same function, i.e., $f(x; \theta) = f(x; \theta')$. This transformation corresponds to permuting the hidden neurons and defines the standard symmetry of the parametrization.

Now, consider a degenerate parameter tuple $\theta_0 = (W_1, W_2)$ with only one active neuron given by

$$W_1 = \begin{bmatrix} 1 & 1 \\ 2 & 3 \end{bmatrix}, \qquad W_2 = \begin{bmatrix} -2 & 0 \end{bmatrix}, \tag{18}$$

The resulting end-to-end function is $f(x; \theta_0) = -2\sigma(x_1 + x_2)$ and has a larger symmetry group than $K_1$ due to the existence of an inactive neuron. Moreover, this function is invariant under the group $G = S_2$ acting on the input space $V_0 = \mathbb{R}^2$ by coordinate permutation, as the sum $x_1 + x_2$ is symmetric. However, the first layer map $f_1(x; \theta_1) = \sigma(W_1x)$ is not equivariant with respect to this action, because the second row $\begin{bmatrix} 2 & 3 \end{bmatrix}$ of $W_1$ breaks the input symmetry.

Although $\theta_0$ is not weakly identifiable (otherwise $W_1$ should have been equivariant under $S_2$ via Theorem 4.1), there exists another parameter tuple $\theta_0' = (W_1', W_2')$ defined by

$$W_1' = \begin{bmatrix} 1 & 1 \\ 0 & 0 \end{bmatrix}, \qquad W_2' = \begin{bmatrix} -2 & 0 \end{bmatrix}, \tag{19}$$

such that $f(x; \theta_0) = f(x; \theta_0')$. In contrary to $\theta_0$, we claim that $\theta_0'$ is weakly identifiable. To see this, we show that $\theta_0'$ corresponds to a submodel as the following diagrams commute

$$
\begin{array}{ccc}
V_0 \times \Theta_1 \ni \left( \begin{bmatrix} x_1 \\ x_2 \end{bmatrix}, \begin{bmatrix} 1 & 1 \\ 0 & 0 \end{bmatrix} \right) \xrightarrow{\ f_1\ } V_1 \ni \begin{bmatrix} \sigma(x_1 + x_2) \\ 0 \end{bmatrix} & & \\
\big\uparrow {\scriptstyle \mathrm{Id} \times \beta_1} & & \\
V_0 \times \widetilde{\Theta}_1 \ni \left( \begin{bmatrix} x_1 \\ x_2 \end{bmatrix}, \begin{bmatrix} 1 & 1 \end{bmatrix} \right) & & \\
\big\downarrow {\scriptstyle \alpha_0^* \times \mathrm{Id}} & & \\
\widetilde{V}_0 \times \widetilde{\Theta}_1 \ni \left( \begin{bmatrix} x_1 \\ x_2 \end{bmatrix}, \begin{bmatrix} 1 & 1 \end{bmatrix} \right) \xrightarrow{\ \widetilde{f}_1\ } \widetilde{V}_1 \ni \sigma(x_1 + x_2) & & (20)
\end{array}
$$

$$
\begin{array}{ccc}
V_1 \times \Theta_2 \ni \left( \begin{bmatrix} z_1 \\ z_2 \end{bmatrix}, \begin{bmatrix} -2 & 0 \end{bmatrix} \right) \xrightarrow{\ f_2\ } V_2 \ni -2\sigma(z_1) \\
\big\uparrow {\scriptstyle \mathrm{Id} \times \beta_2} \\
V_1 \times \widetilde{\Theta}_2 \ni \left( \begin{bmatrix} z_1 \\ z_2 \end{bmatrix}, -2 \right) \\
\big\downarrow {\scriptstyle \alpha_1^* \times \mathrm{Id}} \\
\widetilde{V}_1 \times \widetilde{\Theta}_2 \ni (z_1, -2) \xrightarrow{\ \widetilde{f}_2\ } \widetilde{V}_2 \ni -2\sigma(z_1) \\
(21)
\end{array}
$$

To this end, we remark that the weight matrix $W_1'$ satisfies $W_1'g = W_1'$ for all $g \in S_2$. This implies that the first layer maps the standard permutation representation on $V_0$ to the trivial representation on the latent space. Consequently, the second layer map parameterized by $W_2'$ is automatically equivariant between two trivial representations and thus both layers are $G$-equivariant, which is compatible with our statement in Section 4.

## C. Experimental Details

In this section we provide more details on the experiments presented in Section 6. All experiments were carried out on a single NVIDIA L4 GPU. The hyperparameters selected below were chosen by hand to obtain reasonable training results. The autoencoder and classification performances of our networks are modest, but the networks serve well for qualitative illustration of the theoretical results.

We train with batch size 4096 for 1000 epochs with initial learning rate 0.001 decaying according to a cosine schedule. A large weight decay value of 0.99 is chosen to increase the distinctiveness of the visualized filters. We train with data augmentation consisting of left-right mirroring and standard photometric alterations such as color jitter. Further, we use random crops of the images, resized to size $14 \times 14$. We use a combined loss function of the form

$$\ell(f(x), y) + \lambda \cdot \text{MSE}\left(f(x), \texttt{mirror}[f(\texttt{mirror}[x])]\right) \tag{22}$$

where $\ell$ is mean-squared error (MSE) loss and $y = x$ in the autoencoder experiments and $\ell$ is cross-entropy loss and $y$ consists of class labels in the classification experiments. In the classification experiments, the outer `mirror` is not used (as class labels are mirror invariant). $\lambda$ is 0 for the first half of the training epochs and then set to 5 for the autoencoder experiments and 1 for the classification experiments. The networks used consist of four MLP layers in the MLP experiments and one self-attention layer followed by three MLP layers in the self-attention experiment. The internal feature dimension of all MLP layers except the first is 256, in the first layer we use feature dimension 64 for the MLP experiments and feature dimension 256 divided into 8 heads for the self-attention experiment.

