# OpenReview forum: "Identifiable Equivariant Networks are Layerwise Equivariant"
_ICML.cc/2026/Conference — ICML 2026 regular_

### Official Review · Reviewer_9K8t · 2026-02-27

**Soundness:** 3
**Presentation:** 3
**Significance:** 2
**Originality:** 3
**Overall Recommendation:** 4
**Confidence:** 3

**Summary:**

This paper has proved that if a neural network is end-to-end equivariant and (weakly) identifiable, each hidden layer of this network must also be equivariant.
The proposed theorem explains why unconstrained networks can automatically learn symmetries during training.
Experiments have shown how unconstrained models naturally rearrange their neurons or attention heads to handle flipped images.

**Compliance With Llm Reviewing Policy:**

Affirmed.

**Final Justification:**

I appreciate the authors’ rebuttal. I maintain my assessment of the paper and keep my original recommendation.

**Key Questions For Authors:**

see weaknesses

**Limitations:**

yes

**Strengths And Weaknesses:**

## Strengths

1. The paper have proposes an interesting theory that under certain conditions, an end-to-end equivariant network must be layerwise equivariant, which provides a theoretical view for explaining the phenomenon that models can automatically learn symmetry from data.

2. This paper bridges geometric deep learning with parameter identifiability, which opens a new way to understand the structure of equivariant networks.


## Weaknesses

1. Another answer to the question "how end-to-end equivariance can be engineered into a deep network without a layerwise approach" is the canonicalization network[1], which disentangles the network's equivariant part and non-equivariant part by inserting a learnable canonicalization network to canonicalize the input and the output and thus free the learning of equivariance in the middle layer.
This paper lacks a discussion of this kind of end-to-end equivariant networks.

2. The experiments only provide qualitative results. To better demonstrate the learned layerwise equivariance, quantitative results such as the equivariance error (EE) of each hidden layer could be reported.


[1] Kaba, Sékou-Oumar, et al. "Equivariance with learned canonicalization functions." International Conference on Machine Learning. PMLR, 2023.

---

> ### Author Rebuttal · Authors · 2026-03-30
>
> We thank the reviewer for the review and for raising two interesting points, which we discuss below.
>
> **Canonicalization**
>
> A canonicalization network $(h(x))^{-1} \cdot f(h(x)\cdot x)$ can still be viewed as layerwise equivariant, since the canonicalization function $h$ itself is equivariant (and hence layerwise equivariant if it is identifiable) and the latent spaces inside $f$ are invariant. We are not aware of identifiability results for networks with this specific multiplicative structure, where $h$ outputs a transformation used to modify $x$ before applying $f$. But if such identifiability results are developed, our theory directly provides corresponding results on layerwise equivariance for unconstrained networks of the form $(h(x))^{-1} \cdot f(h(x)\cdot x)$. We thank the reviewer for the interesting comment.
>
> **Quantitative results**
>
> As suggested by the reviewer, we have conducted a quantitative study for the MLPs in Figure 2 as follows. We feed 100k noise images $x$ and their mirrored versions $\tilde x$ through a couple of layers of the network to obtain $f(x)$ and $f(\tilde x)$. We then estimate a linear transformation $A$ such that $A f(x)\approx f(\tilde x)$ by least squares estimation. We feed 100k new noise images $y$ and their mirrored versions $\tilde y$ through $f$. Finally, we compute the relative error as $\|f(\tilde y) - A f(y) \| / \|f(y)\|$, which is a measure of how (linearly) equivariant $f$ is. We report the median of this error in the table below.
> We include both left-right mirroring, which is what the networks are trained to be equivariant to, and up-down mirroring as a reference. $f_1$ means the first linear layer and the first nonlinearity. $f_2$ means the first linear layer, the first nonlinearity and the second linear layer. The quantitative results are aligned with the discussion in the paper. Tanh-networks are equivariant both for $f_1$ and $f_2$, whereas GELU networks are much more equivariant for $f_2$. This can be explained by the fact that the GELU networks make use of the discussed identity $\mathrm{GELU}(x) - \mathrm{GELU}(-x) = x$ which requires two linear layers to realize. Left-right equivariance is stronger than up-down equivariance in all cases, although the autoencoders do exhibit a degree of up-down equivariance even without being trained for it. This can be explained by the fact that the autoencoding task in itself is equivariant to up-down mirroring, whereas this is more unclear for classification.
>
> |       | Mirroring  | Autoencoder (Tanh) | Classifier (Tanh) | Autoencoder (GELU) | Classifier (GELU) |
> |-------|------------|--------------------|-------------------|--------------------|-------------------|
> | $f_1$ | left-right | 0.029              | 0.077             | 0.40               | 0.19              |
> | $f_1$ | up-down    | 0.15               | 0.48              | 0.49               | 0.56              |
> | $f_2$ | left-right | 0.022              | 0.054             | 0.11               | 0.064             |
> | $f_2$ | up-down    | 0.13               | 0.38              | 0.18               | 0.42              |
>
>
> In the updated version of the paper, we will include these results in Sec. 6.

---

> > ### Author Rebuttal · Reviewer_9K8t · 2026-04-02
> >
> > I appreciate the authors’ rebuttal. I maintain my assessment of the paper and keep my original recommendation.

---

### Official Review · Reviewer_jcmB · 2026-03-09

**Soundness:** 3
**Presentation:** 3
**Significance:** 3
**Originality:** 4
**Overall Recommendation:** 5
**Confidence:** 4

**Summary:**

This paper proposes that traditional equivariant neural networks are usually constructed by stacking, but in fact, any equivariant neural network can be understood as a stack of several layers of equivariant latent space layers.

**Compliance With Llm Reviewing Policy:**

Affirmed.

**Final Justification:**

See rebuttal acknowledgement.

**Key Questions For Authors:**

See above.

**Limitations:**

Yes.

**Strengths And Weaknesses:**

The strength of this paper lies in its powerful conclusions, which, to my knowledge, have not been reached before. However, the following shortcomings and questions exist regarding these conclusions.
Based on the author's reference list, I believe the scope of the author's discussion should include common groups, such as permutation groups and Euclidean groups, but it seems that the discussion in the article only applies to a small range of groups such as permutation groups and $p4m$ groups. It seems that it is not applicable when considering Euclidean groups $E(3)$. Here are two simple examples:
- In Def. 3.2, why does map pair $(\alpha\_i, \alpha\_i\^\*)$ exist? In Euclidean groups $E(3)$, [A] shows that, with input and output space given, any function (models with $\theta$) on symmetric graphs (data $x$) will degenerate to zero vector, and there exist no map pair $(\alpha\_i, \alpha\_i\^\*)$ here since bijection is broken.
- In Eq. (5), how can the rotation of translation be separated and applied to the parameters?

Furthermore, some statements are unreasonable or lack supporting references.
- In Line 36 right "In other words, approximately equivariant neural networks are often approximately layerwise equivariant...", I am not aware of any literature that has reached this conclusion, and based on my own experience, this assertion is incorrect. If the author has relevant references, please list them and explain under what circumstances they hold true.
-  In line 112 left, it is recommended to add references such as [B, C].

In summary, if it truly is a matter of application scope, I believe simple text modifications would suffice (though my understanding might be incorrect). Furthermore, I suggest the authors discuss more deeply how their conclusions will guide model architecture design and their practical implications.

If the above issues are resolved, I am willing to increase my score.

[A] Are High-Degree Representations Really Unnecessary in Equivarinat Graph Neural Networks?

[B] On the Universality of Rotation Equivariant Point Cloud Networks

[C] Universally Invariant Learning in Equivariant GNNs

---

> ### Author Rebuttal · Authors · 2026-03-30
>
> We thank the reviewer for the review and for their interest in our results. We hope to clarify the raised weaknesses below.
>
> Let us first address the concerns regarding the Euclidean group. The theory presented in this paper works for any group, including the Euclidean group. In this case, Theorem 4.1 implies that for an identifiable network that is equivariant under the Euclidean group, the group must act on each latent space such that each layer is equivariant.
>
> A caveat is that, at present, identifiability results are known for MLPs with pointwise activation functions, limiting the latent group actions to be (scaled) permutations. Since the Euclidean group has no non-trivial finite permutation representations, this severely limits what equivariant functions can be obtained with MLPs with pointwise activation functions. There are two interesting points to be made here. Firstly, equivariance under a finite subgroup of the Euclidean group is still possible with pointwise activation functions and this is how for instance group equivariant convolutional networks (Cohen & Welling, ICML 2016) are constructed. Secondly, a standard way to design networks that are equivariant under the Euclidean group is via non-pointwise activation functions  – for instance norm-based activation functions or the directional ReLU in Vector Neurons (Deng & al., ICCV 2021). Our theorem works in this setting as well, with the caveat that identifiability results for MLPs with non-pointwise activation functions are not known in the literature. Once they are proven, our theory applies. We believe that identifiability for non-pointwise activations is an interesting direction for future research.
>
> Regarding the specific points raised in the review:
>
>  - Even if the function defined by the model vanishes, the pair $(\alpha_i, \alpha_i^\*)$ always exists. Indeed, in that case, the submodel’s latent spaces will degenerate to the zero vector space  $\widetilde{V}_i = \{ 0\}$, and both $\alpha_i$ and $\alpha_i^\*$ will be zero linear maps. Note that the second condition in Definition 3.2 does not imply that $\alpha_i$ is bijective, but only that it is injective, which is satisfied in this case. In conclusion, we believe that the arguments in [A] do not contradict our theory.
>
> - In a linear layer $(W, b)$ applied to 3D coordinates $x$, we have that $W(Rx + t) +b = (WR)x + (Wt + b)$, so that a roto-translation $(R, t)$ can be fused into the parameters $(W, b)$. The action $g^{-1}\cdot$ on the parameters, which appears in Eq. 5, is $(R, t)^{-1}\cdot (W, b) = (WR, Wt + b)$. Thus, the adjunction property holds.
>
> - In our understanding, some works have indeed observed this phenomenon, empirically.  In fact, in the sentence before line 36, we have cited several papers with empirical studies reaching the conclusion that equivariant networks are often layerwise equivariant. This is also what we observe in our experiments in Section 6; the trained networks are only approximately equivariant, and their weights are close to equivariant matrices.
>
> - We agree that the mentioned references are relevant, especially with regard to the Euclidean group. We will incorporate them in line 112.
>
> Lastly, regarding practical implications, we believe that this line of research might help, in the long run, to map out the landscape of equivariant networks. Specifically, for practitioners, Theorem 4.1 implies that when choosing an identifiable architecture, there are no “hidden” equivariant models that can not be obtained by building the network from equivariant layers. We will add this discussion to the introduction.

---

> > ### Author Rebuttal · Reviewer_jcmB · 2026-04-02
> >
> > I appreciate the authors' replies, but they haven't answered my questions yet. My main concerns here are the existence of mapping pairs and three-dimensional equivariant mappings. Regarding the former, I believe the authors' explanation is incorrect; and regarding the latter, I think there might be some misunderstanding.
> >
> > Regarding the first point, the existence of mapping pairs, the authors explain that Def. 3.2 only requires injectivity, which is indeed true. However, [A] mentions that degeneration can indeed break injectivity (mapping multiple different inputs to the same all-zero output), therefore, $\alpha\^\*$ cannot be found such that $\alpha\^*\circ\alpha=\text{Id}$. This is indeed a tricky problem. I designed a patch combining [B] and a follow-up work [D] to [A], which might solve this issue. The statement mentioned in [A] is actually due to a poorly chosen representation space, specifically $V_i,\tilde{V}_i$ (in technical terms, not obtaining the corresponding degree [D]). From [B,D], we know that ideally, obtaining a sufficiently high degree or a sufficient representation space can guarantee the existence of equivariant mappings for the corresponding injective (or even bijective) mappings. From this perspective, the authors need to modify the description in Def. 3.2, considering an ideal representation space that allows mapping pairs to exist. While this might weaken the existing theoretical results, I believe this won't affect the article's significance or practical guiding value. The conclusion in [C] demonstrates that for asymmetric cases, a first-degree (Cartesian coordinate space) bijection can exist, while for symmetric cases in [A], in reality, appropriate destruction of equivariance [E] is generally considered for handling (this [E] is, to some extent, a follow-up to [1] mentioned by Reviewer 9K8t; I suggest the article could dedicate a separate section to discussing these).
> >
> > Furthermore, regarding 3D equivariant mappings, in equivariant graph neural networks, coordinates are generally not used as input to the MLP; the more common form is $\texttt{MLP}(\boldsymbol{m}\_{ij})\cdot(\vec{\boldsymbol{x}}\_{i}-\vec{\boldsymbol{x}}\_{j})$, generally only processing scalars. Therefore, I'm unsure if there was some misunderstanding in my discussion with the authors regarding this section; at least I believe this rotation is inseparable. Of course, removing this section wouldn't affect the article's value.
> >
> > [D] Reducing Symmetry Increase in Equivariant Neural Networks
> >
> > [E] Improving equivariant networks with probabilistic symmetry breaking
> >
> > ---
> >
> > Additional comments for “Reply Rebuttal Comment by Authors”
> >
> > Thank you for the reply; this discussion has also been very beneficial to me. I believe the article is now logically consistent, and I recommend a "clear accept."

---

> > > ### Author Response · Authors · 2026-04-03
> > >
> > > We thank the reviewer for acknowledging the rebuttal. We believe that the two remaining points are due to misunderstandings, and both solvable. We hope to explain this in the following.
> > >
> > > We believe that our previous reply on Def. 3.2 was imprecise w.r.t. the question asked. The injectivity criterion in Def. 3.2 is for the map $\alpha_i : \widetilde{V}_i \to V_i$ between the submodel’s latent space, and the original model’s latent space. It does not refer to the layer map $x \mapsto f_i(x; \theta_i)$, which is not required to be injective. Note that the choice of the submodel representation space $\widetilde{V}_i$ is part of the structure of the submodel, and it can be any set. This freedom enables the pair $(\alpha_i, \alpha_i^*)$ to exist in broad settings. For example, in case $f_i$ collapses to the zero map, such as in [A], then the definition can still cover this by setting trivial submodel latent space $\widetilde{V}_i = \{0\}$. This is what we tried to convey in the previous reply.
> > >
> > > However, the reviewer may furthermore be concerned that the network layer map $x \mapsto f_i(x; \theta_i)$ itself must always collapse to the zero map in our setting. We wish to stress that Def. 3.2 does not have to be modified to cover the ideal representation spaces mentioned by the reviewer, as we do not have any restrictions on $V_i$ in our setting. Thus, $V_i$ can be equipped with arbitrary high-degree irreps of $SO(3)$ and still fit into the framework. Def 3.2 therefore covers both the case where $f_i$ is the zero map (in which case $\alpha_i$ exists, as mentioned above), and the case where it is not. Yet, we
> > > acknowledge that the case where $f_i$ collapses to the zero map is interesting to discuss separately. We will add a remark on it separately, citing the references [A-E] as examples of when this can be the case.
> > >
> > > Regarding the adjunction property, we thank the reviewer for clarifying that they are asking about the EGNN case. In our previous reply we were indeed considering a general unconstrained MLP. The case when the network is constrained to be layerwise equivariant a priori is a very interesting special case of our theory, which is covered via the generalized adjunction property in Eq. 6. For instance, in EGNNs, if we let $R^{-1}\cdot \theta = \theta$, then Eq. 6 holds since the MLP portion of the layer is invariant, and the $(x_i - x_j)$ is equivariant as mentioned by the reviewer.
> > >
> > > We hope that the above clarifies the remaining concerns.

---

### Official Review · Reviewer_YvYQ · 2026-03-10

**Soundness:** 2
**Presentation:** 2
**Significance:** 3
**Originality:** 2
**Overall Recommendation:** 3
**Confidence:** 3

**Summary:**

This paper proves the following theoretical result: under the identifiablity assumption, equivariance of the network function implies equivariance of the function of each layer. The statement and proof of the result are presented in a highly abstract manner. For concrete networks, including MLPs and attention nets, the authors discuss how the abstract result, as well as the conditions necessary for it to hold, actually instantiate.

**Compliance With Llm Reviewing Policy:**

Affirmed.

**Key Questions For Authors:**

1. Regarding the group actions with respect to the equivariance, what is the connection between the group actions on the layers and those on the input/output spaces?




2. Identifiability is sufficient for the claimed result to hold; is it also a necessary condition?

**Limitations:**

Yes.

**Strengths And Weaknesses:**

Strength:
1. The notions of equivariance, symmetries, identifiablity, are defined formally.

2. The research question, what are the connections between equivariance of the network function and that of each layer, is well-motivated in Sec.1.

3. One example is provided in the appendix to help understand the abstract definitions.


Weakness:
1. The connection between the group actions on the layers and the group actions on the input/output space, which is part of the central question in this paper, is not discussed carefully and in detail. When it comes to the connection, the sentences become not so clear. For example, in the abstract, “… end-to-end function .. with respect to group actions, … layers are equivariant with respect to some group actions…”. What do the “group actions” and “some group actions” refer to? Are they the same? Such vagueness is not properly handled in the main body after the abstract. Even Thm 4.1, which is their main result, is not clearly stated: “Then there exist group actions by G on Vi, such that f_i is G-equivariant at θ_i.” According to the definition of G-equivariance in Def. 3.6, f_i should be equivariant to all actions in G, why “there exist group actions on G”.

2. Identifiability is the most important assumption for the main result to hold. The authors discussed existing work on identifiability in MLPs and attention nets in Sec. 5. The statement regarding identifiability in MLPs, however, is misleading for the following reason. The identifiability has been proven for some activations, as listed by the authors, but is also disproven for other activations, such as ReLU [1] [2]. Conjecture 5.1, which states that the (weak) identifiability holds in general, is not correct per se. Both Conjecture 5.1 and Conjecture 5.3 (a corollary of Conjecture 5.1), use the premise “under mild conditions”, but the authors did not explain what they mean by “mild conditions”.

3. It is not so clear to me what the message is in Sec. 6-experiment. Is the expeirment result the following: network equivariance leads to layer equivariance, as claimed by the theory? But the theory holds under the identifiability assumption, does this assumption hold in the experiment?

[1] arXiv:2209.04036

[2] arXiv:2306.06179

---

> ### Author Rebuttal · Authors · 2026-03-30
>
> We thank the reviewer for the review and for writing detailed weaknesses and questions which we hope to address below.
>
> **Weaknesses**
>
> 1. We believe that the confusion stems from different interpretations of the term “group action”. By “group action”, we refer to a symmetry structure over a set $X$ – more technically, a map $\pi  \colon G \times X \rightarrow X$. Thus, “a group action” is not a specific group element $g\in G$, but the whole map $\pi$ specifying how each $g\in G$ transforms elements of the set $X$. This is standard terminology in group theory. Note that we use the notation $g\cdot x$ to denote $\pi(g, x)$. This notation is standard but overloaded, since it omits the map $\pi$; it can refer to different group actions, depending on the context. Now, for our main result we fix two (potentially distinct) actions on the input and output space, such that equivariance (Def. 3.6) holds for all $g \in G$. Theorem 4.1 says that there exist group actions on all latent spaces, such that Def. 3.6 holds for all layers, again for all $g\in G$. These latent group actions can be different (and often are, in practice). This is what we tried to convey in the abstract; the adjective “some” refers to the fact that the latent actions are guaranteed to exist; all equivariance properties hold for all group elements.  We acknowledge that the terminology and notation might lead to confusion. In the updated version of the paper, we will amend this by including a definition of group action (which is now missing from the paper), and by rephrasing the abstract to make it clearer.
>
>
>
> 2. We acknowledge that the formulation of Conjecture 5.1 is not completely precise. By “mild conditions”, we had in mind the following fact. In [A], identifiability is proved for a family of meromorphic activations that is dense, with respect to the sup norm, in the class of piecewise $C^1$ functions whose derivative has bounded variation. This is the sense in which we used the term “mild condition”, as it holds for a dense family in a natural function class. For ReLU, however, the situation is subtle, and the reviewer is right to point this out. We believe that Conjecture 5.1 might hold for ReLU networks, and that it is, in principle, compatible with the works [1,2] mentioned in the review. The caveat is that, for ReLU networks, we believe that the symmetry groups $K_i$ need to be chosen larger than permutations and rescalling a priori, as our theory allows. The notion of submodel will also change, and a large class of parameters will lay in the image of submodels. Intuitively, this is due to the piecewise-linear nature of ReLU networks, that leads to non-trivial linear submodels. Thus, our notion of weak identifiability is more flexible than it might seem. As [1,2] demonstrate, understanding this picture for ReLU networks is extremely challenging, and represents an important direction for future work. In the updated version of the paper, we will expand the discussion in lines 275-278, and relate it to the results in [1,2].
>
>
>
> 3. The intended message of our empirical investigation is twofold. First, to confirm that the theory actually holds in practice. This is not automatic, since the theory assumes perfectly equivariant models, while trained networks will be only approximately equivariant. Second, we indeed wanted to experiment with cases where the identifiability is not understood. While for Tanh identifiability is proven, for GELU it is not, and it is unclear what type of group actions are expected. The experiments highlight the difference between these two cases. For Tanh, identifiability means that group equivariance implies that signed-permutation actions exist on the latent spaces, as we observe in Figure 2. For GELU, we do not get signed permutation actions, and the question about what the correct notion of layerwise equivariance remains open, along with the identifiability question. We agree that this subtlety is worth highlighting, and we will discuss it in the updated version of the paper.
>
> [A] arXiv: 1906.06994
>
> **Questions**
>
> 1. Please refer to the answer to weakness 1 above.
>
>
> 2. We believe that this is an interesting theoretical question, to which we do not have a complete answer. Identifiability might be a necessary assumption, but we did not manage to prove it mathematically. We will, however, add a remark on this in the paper, mentioning it as an interesting question for future investigation.

---

> > ### Author Rebuttal · Reviewer_YvYQ · 2026-04-03
> >
> > Thank you for the rebuttal.

---

### Official Review · Reviewer_DJSK · 2026-03-12

**Soundness:** 4
**Presentation:** 4
**Significance:** 3
**Originality:** 3
**Overall Recommendation:** 5
**Confidence:** 4

**Summary:**

The paper proves that generic neural networks (intended as in the usual sense of reiterated composition of affine maps and entry-wise activations) which are equivariant functions can be reparameterized as layer-wise equivariant functions, namely, neural networks where affine layers are equivariant by themselves.

**Compliance With Llm Reviewing Policy:**

Affirmed.

**Final Justification:**

The paper tackles a relevant problem at the essence of representation learning showing the existence of symmetrical representations under certain identifiability and adjunction conditions.

As I have stated in my review, the paper is well-presented, sound and well-positioned in the literature.

My main concearns regarding accessibility of the paper have been addressed during the rebuttal phase: the abstract nature of the paper is an advantage in terms of generality and compactness of presentation but those tool may *erroneously* feel distant from the current research objectives. Adding adequate examples would solve this issue.

Similarly, the authors ensure that the limitations of the assumptions made will be better circumscribed.
For this reasons, I confirm my positive score.

**Key Questions For Authors:**

The main proof of the theorem hinges around the adjunction property in equation (5) and (6). Can the authors show examples of failures of  this property?

**Limitations:**

yes

**Strengths And Weaknesses:**

**Strengths:** \
The paper is well-presented, mathematically sound, of great interest for the geometric deep learning community and well-positioned with respect to the corresponding literature.

**Weaknesses:**
* The paper could be better positioned with respect to the identifiability literature. The paper presents the concept of submodel and weak-identifiability and illustrates it through the example presented in Appendix B. However, this concept may be better discussed with regard to elements already present in the identifiability literature such as the *no-clones condition* or *non degeneracy* from [1].
* The experimental part in Section 6 is interesting and well-presented. However, it may be relevant to underline that the shown models are only (I suppose) approximately equivariant, which is not exactly the presented framework. Moreover, the presented theory shows an existential property, i.e., there exist a layer-wise equivariant parameterization, giving no guidelines on how such parameterization could emerge through training in an unconstrained model.
This experimental part shows that this parameterization is also obtained after optimization, which is a relevant venue for future work that chould be highlighted.

**References:** \
[1] Vlacic and Bölcskei, *Neural Network Identiﬁability for a Family of Sigmoidal Nonlinearities*, 2021

---

> ### Author Rebuttal · Authors · 2026-03-30
>
> We thank the reviewer for the review and for the appreciation. We wish to comment on the weaknesses raised:
>
>
> - We agree that the connection with prior identifiability literature could be made more explicit. Indeed, the no-clones and non-degeneracy conditions from [1] correspond, in our language, to parameters that do not come from submodels. Thus, they align perfectly (and actually motivate) our notion of weak identifiability. This is explained, to an extent, in the “Submodels” paragraph of Sec. 5.1, albeit with a different terminology: our “inactive” and “redundant” neurons correspond to “cloned” and “degenerate” neurons from [1], respectively. We will clarify this in the updated version of the paper.
>
>
> - The fact that, in practice, networks are only approximately equivariant is an important point. It was a core motivation for performing the empirical analysis, showing that the theory (which assumes perfect equivariance) still applies in more relaxed and noisy conditions. We will highlight this motivation at the beginning of Sec. 6. Moreover, how equivariance emerges during the training process is indeed a crucial question, which is likely to be extremely challenging, mathematically. We will mention it in Sec. 7, as suggested.
>
> Let us also comment on the question raised around the adjunction property. In our view, adjunction is a very natural property. It is, in fact, satisfied by most of the standard architectures. It is still possible to find examples where the adjunction property is not satisfied, in particular if the weights of the first layer are significantly constrained. For instance, if the rows of the first weight matrix are constrained to have unitary norm, then the adjunction property does not hold for non-orthogonal group actions on the input. Moreover, if the first layer is constrained to be equivariant under specific group actions on the input and output, then for other group actions on the input, the adjunction property may fail to hold. A concrete example of this is the DeepSets architecture, which is permutation equivariant. In this case, the linear weight has the form $W = a I + b \mathbf{1} \mathbf{1}^T$, which can not “absorb” an arbitrary group action, as required by the adjunction property (it does however absorb permutations in the sense of Eq. 6).

---

> > ### Author Rebuttal · Reviewer_DJSK · 2026-04-02
> >
> > I sincerly thank the authors for their answers which elucidated a large portion of concerns raised in the review and convinced me to confirm my positive assessement.
> >
> > In particular, I want to thank the authors for their clear and honest answer regarding the adjunction property which I would like to explore further. DeepSets is indeed the example that I had in mind when I wrote the quesiton. If I am not wrong, I would also take into account Invariant Graph Networks [1] as a relevant example of failure for this property and which covers a wide range of equivariant models such as graph neural networks.
> >
> > For this reason, I would soften the following statement in your reply and I suggest to clarify this aspect in the manuscript:
> >
> > > [...] adjunction is a very natural property. It is, in fact, satisfied by most of the standard architectures.
> >
> > I would also encourage to update the manuscript with examples showing the failure of this property to help the reader to understand the domain of applicability of the presented framework.
> > It would also be valuable to show how Theorem 4.1 would fail in these circumstances; if it does.
> >
> > [1] Maron et al., *Invariant and Equivariant Graph Networks*, ICLR 2019

---

> > > ### Author Response · Authors · 2026-04-03
> > >
> > > We thank the reviewer for acknowledging our rebuttal.
> > >
> > > We agree that the statement about adjunction should be softened in the revised manuscript. In particular, we will clarify that adjunction is a natural assumption in several known models, but not a universal one. We will add the discussion of DeepSets and Invariant Graph Networks [1] to illustrate that the property can fail, for example, when the first or last layer is subject to architectural constraints.
> > >
> > > Regarding Theorem 4.1, our current proof does not hold without the adjunction property, and we do not know whether the statement is true. We plan to investigate it in the future.

---

### Decision · Program_Chairs · 2026-04-30

**Decision:**

Accept (regular)

**Comment:**

The authors show that if an equivariant network is identifiable, then it can be realized with equivariant layers. In other words, there is no benefit, in terms of expressive power, in going beyond layer wise equivariant networks to end-to-end equivariant networks, under the identifiability condition.

The paper has a clean theoretical message, and I believe it is of interest to the ICML community, specifically GeoML researchers. It is also very well presented and well positioned, according to Reviewer DJSK:

> Reviewer DJSK: The paper tackles a relevant problem at the essence of representation learning showing the existence of symmetrical representations under certain identifiability and adjunction conditions.

> Reviewer DJSK: As I have stated in my review, the paper is well-presented, sound and well-positioned in the literature.

Given that the reviewers all agree that the results are interesting and relevant, I recommend acceptance.

Note: Reviewer YvYQ provided a number of concerns that are already addressed in the rebuttal. I want to ask the authors to provide this explanation in the next version of the paper, as it looks essential to have it in the paper.